# OpenReview forum: "Measuring and Guiding Monosemanticity"
_NeurIPS.cc/2025/Conference — NeurIPS 2025 spotlight_

### Official Review · Reviewer_cJ89 · 2025-06-22

**Clarity:** 3
**Significance:** 3
**Originality:** 2
**Rating:** 5
**Confidence:** 4

**Summary:**

This paper opens by introducing new metrics to measure two aspects of features in a language model: disentanglement, or how few features are necessary to encode a given concept, and monosemanticity, or how few concepts are encoded by a given feature. Ideally, the features output by a method should be both disentangled and monosemantic. It adds to a growing body of work in showing that SAEs are neither disentangled (enough) nor monosemantic (enough). The paper then introduces G-SAEs, a variant of SAEs that add a supervised term to the loss to target only the features that are known to be present in the text. Using the new metrics, the paper finds that G-SAEs are considerably more monosemantic and disentangled than SAEs.

**Questions:**

- Can you provide any extraneous evaluation of your new metric and/or the G-SAEs that does not depend upon each other? That is, some evaluation in terms of already accepted norms to provide more confidence in the soundness of these contributions.
- Have you thought of how G-SAEs may be trained in an unsupervised fashion? I know that this is not the aim of the paper, and I don’t necessarily expect new results, but I would like to know if there are ways to extend G-SAEs to a general setting.
- Your new metrics apply both to SAEs and to other forms of features that are used in interpretability, e.g., circuits. It would be interesting (and also support point 1 above) if you could use your metrics to compare, e.g., SAEs and transformer circuits in terms of how disentangled they are. My point is that your metrics may be more generally useful to compare not individual feature extractors but classes of feature extractors.

**Ethical Concerns:**

["NO or VERY MINOR ethics concerns only"]

**Final Justification:**

I believe that the authors have sufficiently addressed my concerns. Specifically, the newly introduced results on (1) using HM v/s AM for computing scores (2) Connection to conventional evaluation practices (3) the clarification of Figure 4 address the key concerns I had. Therefore, I am raising my score to "5: Accept" as I believe that in its current form, this paper will prove of interest to several researchers and practitioners.

**Limitations:**

Recently, SAEs have come under fire for being “bad” for various reasons: they don’t always find all the features that a model represents [1], may learn different features when trained with different seeds or perturbed weights [2], and are not very monosemantic [3]. The paper does acknowledge these shortcomings at various isolated places, and to be clear: I believe that this work is valuable despite the apparent shortcomings of SAEs. Even then, I believe the paper should centrally acknowledge that these are very real drawbacks of SAEs and related families of methods—possibly in a limitations section.

[1] Leask et. al., “Sparse Autoencoders Do Not Find Canonical Units of Analysis,” ICLR 2025

[2] Li et. al., “Interpretability Illusions with Sparse Autoencoders: Evaluating Robustness of Concept Representations,” arXiv preprint arXiv:2505.16004

[3] Huang et. al., “RAVEL: Evaluating Interpretability Methods on Disentangling Language Model Representations,” ACL 2024

**Quality:**

3

**Strengths And Weaknesses:**

**Strengths**

- The study of SAEs is timely, if a bit overcrowded at the moment. I was initially surprised to read that monosemanticity has not been studied thoroughly for SAEs. While I believe that some prior works have indeed studied it to an extent, I think that this paper does a good job of adding to that literature.
- The introduced metrics make sense to me, although they are a bit arbitrary (see Weaknesses below). The paper’s equations are clear without drowning in notation, and Figure 2 does a good job of conveying the design of G-SAEs. The paper is largely easy to read and understand, which is a big plus.
- The evaluations support the claims of the paper, and the edge of G-SAEs well.
- Although I might have missed some, I think the paper cites the most important papers that I know in this space. The authors have done a good job of surveying existing literature.

**Weaknesses**

I will structure this into two parts. First, the weaknesses of the content and contributions of the paper.

- I found the formulations of disentanglement reasonable, but I have some disagreements with Equation 3, the formula for the overall FMS. Specifically,
- I think one should take the harmonic and not the arithmetic mean of the local and global FMS scores to be robust to outliers.
- In my view, both the local and global FMS scores measure entanglement (as the authors probably agree, given by the naming convention). On the other hand, I believe that if we wish to use a single number, it should also measure monosemanticity, or “does this feature measure a single concept?” To answer this question necessarily involves some sort of evaluation of how well the feature can recover other concepts—that is, I don’t think that a metric that only ever evaluates feature a on concept a (and never features a on concepts b) can capture monosemanticity.
- G-SAEs, while performing well, seem to require supervision, which is a clear downside. Additionally, while I do not disagree with the metrics introduced here (or with the fact that one reason to introduce G-SAEs is to test these metrics), it would be nice to also evaluate with more conventional methods to show that the new metrics do in fact agree/improve upon already existing evaluations (for example, prompting an LM to evaluate how faithful an SAE is).
- In Figure 4: while it is true that the gap between the mean activations of absent and present is higher for G-SAEs, there are also two downsides: (1) the spread of, e.g., the present activations is higher for G-SAEs, and (2) normal SAEs never activative for the “absent” case, but G-SAEs show a small but nonzero activation.

Outside of these, I also have some suggestions for the presentation and flow of the paper. I have not taken these into consideration when providing a score below, and they are simply my opinion, but I still believe these changes will improve the experience of a reader.

- I find the fonts and colors in Figures 1 and 3 a bit dull as well as hard to read. I also think that Figure 4(b) is a bit cluttered.
- I feel that the title “Measuring and Guiding Monosemanticity” is a bit too broad to clearly convey what the paper is about. This had me expecting a bit less from the paper going in. I will admit that Section 3 onwards made me change my mind, and I had an enjoyable time reading the paper, but I think that the authors can make the title and Section 1 a bit more specific to make readers excited about their results.

---

> ### Author Rebuttal · Authors · 2025-07-31
>
> **W1 (Harmonic vs arithmetic mean):** We thank the reviewer for the thoughtful suggestion. We chose the arithmetic mean over the harmonic mean in Eq. (3) for the following reasons:
> Complementarity vs. Conjunction: The harmonic mean penalizes low values heavily, which is useful when both components must be high. However, we treat local and global disentanglement as complementary, not strictly conjunctive — strong performance in one should still be rewarded, even if the other is lower (e.g., due to concept spillover).
> Robustness: The harmonic mean is highly sensitive to small fluctuations, making it unstable when one component is near zero (even due to noise or classifier variance). This sensitivity often led to disproportionate score collapse in our experiments.
> Empirical Evidence: As shown in the table below, both means rank models similarly overall, but the harmonic mean introduces sharper drops and reduced score granularity (e.g., Toxcitity for vanilla SAE FMS@1 from 0.26 to 0.01), making the scores harder to interpret.
> Yet, given the data at hand, we agree that both averaging methods offer useful perspectives, and we now include this analysis in the appendix to clarify our choice.
>
> | Dataset | Model | FMS@1 (arith) | FMS@1 (harm) | FMS@5 (arith) | FMS@5 (harm) |
> |--|--|--|--|--|--|
> |Toxicity|Vanilla SAE|0.26|0.01|0.31|0.17|
> | | G-SAE | 0.37 | 0.19 | 0.42 | 0.31 |
> | Shakespeare  | Vanilla SAE | 0.28 | 0.03 | 0.29 | 0.04 |
> | | G-SAE | 0.57 | 0.44 | 0.62 | 0.54 |
> | Privacy | Vanilla SAE  | 0.28 | 0.04 | 0.30 | 0.10 |
> | | G-SAE | 0.62 | 0.51 | 0.65 | 0.58 |
> | **Average**  | Vanilla SAE  | 0.27 | 0.03 | 0.30 | 0.10 |
> | | G-SAE | 0.52 | 0.38 | 0.56 | 0.48 |
>
> **W2 (Measuring Monosemanticity):** We thank the reviewer for this important observation. Monosemanticity indeed requires not just that a feature aligns well with a concept, but that it does not encode others.
> A metric naturally can only measure and capture what is present in the probed data points—namely, the specific boundary-defining concepts found in a labeled dataset. As such, we believe that our approach already incorporates a practical approximation of the general monosemantic idea, as FMS incorporates global and local behavior. The level of granularity can be naturally determined by the selected number and diversity of labeled concepts available in the dataset.
> In addition, when computing feature accuracy (acc_0), we evaluate each feature across all available concepts in the dataset and select the one that separates best, i.e., true positives (correct activations for the target concept) and true negatives (non-activations for other concepts). In this way, each feature is automatically compared against all other concepts of the labeled dataset and only credited for the one it aligns with most strongly.
> We clarified this point in the revised metric description.
>
> **W3 (Supervision):** We don't see this as a weakness as all SAEs require labeled data to construct a steering method. Due to space limitations, please refer to L (Generalizability of G-SAE) of reviewer Agfr.
>
> **W3 (Conventional methods):** Thank you for this suggestion. While we agree that it is important to connect new metrics to established evaluation practices, we note that FMS is designed to capture monosemanticity — a property not addressed by existing metrics. We note that FMS combines the underlying primitive as probes [1,2]: feature-level classification accuracy, and disentanglement measures like Mutual Information Gap [4] used in Variational Autoencoders. Our acc_0 component is conceptually similar to the probing accuracy used in these approaches, but FMS extends this by also measuring exclusivity (local disentanglement) and feature spread (global disentanglement) — aspects not captured by traditional probes. We also observe that acc_0 (higher is better) aligns well with the overall FMS trend and model ranking (c.f. Tab.1), and complementary metrics that assess how pure a split by a feature is, such as Gini impurity (mean of Gini values of second level in Fig 11, the lower is better), yield similar conclusions:
> | Dataset | Model | Gini | acc_0 | FMS@1 |
> |--|--|--|--|--|
> | Toxicity | G-SAE | 0.34 | 0.78  | 0.37  |
> | | Vanilla SAE | 0.38 | 0.69  | 0.26  |
> | Shakespeare | G-SAE | 0.20 | 0.89  | 0.57  |
> | | Vanilla SAE | 0.42 | 0.69  | 0.28  |
>
> Additionally, we confirm that higher FMS scores correspond to improved generation steering (c.f. Tab.2), offering behavioral validation of the metric. However, due to the lack of prior metrics for monosemanticity, a direct comparison or alignment is not straightforward. We therefore see FMS as complementary: extending interpretability analysis into an underexplored axis with practical relevance.
> We appreciate the opportunity to clarify this and have made these relationships more explicit in the revised paper. We hope this addresses your questions.
>
> **W4 (Fig. 4):** Thank you for this careful observation — we agree that Figure 4, in its current form, may give a misleading impression about the activation spread and absence-case behavior of vanilla SAEs. We have clarified the analysis and adjusted the visualization in the revised version:
>
> W4 (Activation spread): The spread difference is primarily due to a lack of clear conception separation of vanilla SAEs, and moreover, showcase their drawback when using them as concept detectors when compared to G-SAE. To provide a fairer, scale-invariant comparison, we report the Ranked Biserial Correlation (RBC) from a Mann–Whitney U test (Section 5.3), which more directly measures separation between "present" and "absent" distributions. This test clearly favors G-SAE, confirming that it more reliably distinguishes the presence of a concept despite wider activation variance. Moreover, note that the plot averages over all concepts, also coarse and sometimes erroneous sentence-label datasets RTP and SP, which naturally introduces noise that cannot be removed and increases the spread. The token-granular PII indeed performs much clearer, as shown in Figure 9.
>
> W4 (absent cases): We apologize for this illustration. We used boxplot settings (25%–75% interquartile range with no outliers), obscuring that vanilla SAEs also produce activations >0 for absent concepts. We've now updated the plot to reflect the 10%–90% range, clearly showing that normal SAEs also activate in the “absent” case. For example, for SP, 16% of the cases activate >0 when actually absent. Note that on the other hand, for the present case, only ~34% activate overall for vanilla SAE, while ~85% for G-SAE. We have updated our figure accordingly and extended our experimental description and discussion.
>
> **Style Feedback:** We appreciate the feedback and improved stylistic appearance of our figures.
>
> **Q1 (Extra Evaluations):** Thank you for this important point. To evaluate the general utility of FMS, we applied it not only to G-SAEs but also to pretrained [3] and our vanilla SAEs. As shown in section 5.2, Table 2, and the Finetuning table seen in L (Generalizability of G-SAE) of reviewer Agfr, FMS scores capture apparent differences in concept disentanglement between vanilla, pretrained, and fine-tuned SAEs. For example, finetuning a pretrained SAE improves FMS@1 from 0.42 to 0.50, while reconstruction performance remains stable. This shows that FMS is model-agnostic and informative across architectures.
>
> To evaluate G-SAE using established tools independent of FMS, we also ran SAE-Bench [5]. As shown in the SAE-Bench results table in Q2 (Cost of monosemanticity) of reviewer QsmM, G-SAE matches vanilla SAEs across all evaluation criteria, while being much more interpretable. Thus demonstrating that the general capabilities of the SAE are retained.
>
> Together, these results demonstrate that:
> - FMS is a meaningful metric even when applied to non-G-SAE models, and
> - G-SAE retains general capabilities, as shown by an evaluation using third-party benchmarks.
>
> **Q2 (Unsupervised training):** We would like to phrase it as automatic supervision, as G-SAE requires labels of some sort. One would need to have a concept labeler that creates labels for G-SAE on the fly. We think this could be partially achieved with current LLMs, but this still needs refinement to achieve fast, reliable labels. Concerning the extension to a more general setting, we conducted a few finetuning runs on a pretrained SAE as described in the answer above, where we demonstrate that a post hoc application of G-SAE is also possible.
>
> **Q3 (General applicability of FMS):** We thank the reviewer and agree that a key strength of FMS is its broad applicability. Though we focus on SAEs here, FMS is model-agnostic and can be applied to other feature extractors like transformer circuits, CLIP neurons, or PCA components. It could also enable comparisons across interpretability methods in terms of concept alignment, disentanglement, and selectivity. While such analysis is beyond the scope of this submission, we are actively exploring it and will highlight FMS’s broader potential in the discussion.
>
> **L1 (SAE critiques):** We thank the reviewer and agree that SAEs have key limitations, such as instability, incomplete feature recovery, and imperfect monosemanticity. We've clarified these in the limitations section. While G-SAE doesn't fully resolve them, its supervision improves concept stability and alignment, mitigating several issues. We’ve revised the paper to emphasize these points more clearly.
>
> [1] Bricken et al., “Towards Monosemanticity: Decomposing Language Models With Dictionary Learning”, Transformer Circuits Thread 2023
>
> [2] Gao, et al., "Scaling and evaluating sparse autoencoders.", ICLR (2025)
>
> [3] Hugginface: EleutherAI/sae-llama-3-8b-32x-v2 @ layer 11
>
> [4] Chen et al., "Isolating sources of disentanglement in variational autoencoders.", NeurIPS (2018)
>
> [5] Karvonen et al., “SAEBench: A Comprehensive Benchmark for Sparse Autoencoders in Language Model Interpretability”, ICML (2025)

---

> > ### Comment · Reviewer_cJ89 · 2025-08-01
> > **Thank you for the rebuttal**
> >
> > I thank the authors for the thorough response. I feel that this rebuttal has sufficiently addressed the most pressing questions I had. In particular, I find the new results on (1) the use of a harmonic v/s arithmetic mean (2) connection to conventional evaluation metrics, and (3) the clarification of Figure 4 meaningful and positive. I am therefore raising my score to "5: Accept."

---

> > > ### Author Response · Authors · 2025-08-03
> > >
> > > We thank the reviewer for the constructive feedback that helped us improve our script-- and the reconsideration of the given score.
> > > If there is anything further we can help with, please let us know.

---

### Official Review · Reviewer_QSmM · 2025-07-02

**Clarity:** 4
**Significance:** 3
**Originality:** 3
**Rating:** 5
**Confidence:** 4

**Summary:**

The paper introduces a new metric, the FMS, to systematically measure how well individual features in a model's latent space correspond to single, interpretable concepts. Building on this, the authors propose G-SAEs, which incorporate labeled concept supervision during training to improve monosemanticity, interpretability, and control in LLMs. Their method presumably enables more precise and reliable model steering which they demonstrate through improved performance in tasks like toxicity detection, writing style identification, and privacy attribute recognition.

**Questions:**

- What is the expected impact of other classifiers used? In the paper, the authors use binary tree classifiers and note that FMS is method-agnostic.
- It would also great to better understand at what cost achieving monosemanticity comes in their proposed G-SAEs.
- In general, a position on the points I raised under “weaknesses”.

**Ethical Concerns:**

["NO or VERY MINOR ethics concerns only"]

**Final Justification:**

Authors addressed my concerns sufficiently during the rebuttal. I recommend to accept the paper to NeurIPS this year as I believe it's a valuable contribution to the field and relevant for the audience at the conference.

**Limitations:**

Yes, although they could say more about potential adversarial or ethical concerns raised through steering concepts (line 339).

**Quality:**

3

**Strengths And Weaknesses:**

##**[EDIT: Authors largely addressed the weaknesses mentioned below in their rebuttal.]**

**Strengths:**
- Overall, the paper presents a meaningful contribution to the interpretability field in my opinion, in particular due to formalizing notions of monosemanticity and providing a generalizable metric for quantifying monosemanticity that is intuitively understandable
- The authors' results are meaningfully interpretable as they’ve also included statistical analysis compared to previous papers I’ve seen on the subject where it was harder to evaluate the actual signal of an evaluation
- The writing in the paper is very clear and accessible

**Weaknesses:**
- My main concern with this paper is that it largely skipped over the initial step in the experimental setup where they state that “given a set of prompts annotated with concept-specific labels” (line 125) and potential impacts of decisions made in this step on the practicality, meaningfulness, and informativeness of their FMS measure and the corresponding G-SAE implementation.
- In particular, many real-world concepts (like “toxicity,” “privacy,” or “style”) are inherently fuzzy, overlapping, or context-dependent in human language. The paper does not sufficiently address how concept granularity, semantic ambiguity, or inter-concept overlap affect the reliability of FMS or the performance of G-SAEs. In addition, while the authors assume access to labeled data, they do not discuss how label quality, annotation bias, or inconsistencies impact their metric or method. This is particularly relevant given that FMS scores and G-SAE performance are sensitive to these labels, as the latent supervision is directly tied to them.
- Relatedly, I would have appreciated more evaluations outside of the ones in the paper that are primarily constrained to three domains (toxicity, Shakespearean style, privacy) to better understand for what concepts this approach works versus where it might fall short (e.g., more abstract, ambiguous concepts).
- In addition, while the writing in general is very clear, I think it could be improved by adding one or two running examples to the writing up until Section 4.

---

> ### Author Rebuttal · Authors · 2025-07-31
>
> We thank the reviewer for the constructive feedback and appreciate that the reviewer finds our paper very clear and accessible.
>
> In the following, we will address your concerns:
>
> **W1 (Addressing labeling):** Label quality is critical in machine learning for both evaluation and training. A labeled dataset is also needed to find concepts in standard SAEs and subsequently use them to detect or steer, making label noise and concept ambiguity not unique to G-SAEs.
>
> In our experiments with RTP and SP, only sample-wise labels are available and indeed several standard tokens are prone to some noisy labels. Nevertheless, the concept detection and steering work well yet naturally with more variance than the token-annotated PII dataset, as demonstrated in Tab. 2. This may be attributed to conflicting annotations averaging out on large enough datasets.
>
> Similarly, our metric is intentionally designed to expose these label ambiguities and inconsistencies; not as a limitation, but as a key feature. All model evaluations require a high quality dataset to be evaluated with, as label errors or coarse granularities skew the usefulness. As such, this issue again is not unique to FMS, but relevant to all machine learning evaluations.
> We have extended our discussion with a paragraph on label quality being a relevant factor for concept supervision, e.g., also in G-SAEs or FMS. Please also see the response below to the related point raised.
>
>
> **W2 (Label diversity):** We appreciate the desire for broad evaluations and agree that understanding the limits of our approach across different concept types is important. Our current choice of domains—privacy (PII) annotated token-grained on 24 different concepts, Shakespearean style (SP) and toxicity (RTP) annotated sentence-wise—was intentional and aimed at capturing a spectrum of concept clarity and ambiguity.
>
> PII serves as an example of well-defined, low-ambiguity concepts: the labels are word-level and often based on strict, rule-based patterns (e.g., email addresses), and G-SAE shows a strong and consistent performance, as shown in Tab. 1 (acc_0).
> In contrast, SP and RTP represent more ambiguous and context-dependent domains. For example, RTP only has sample-level labels, and words like “kill” may appear in both toxic and non-toxic contexts (e.g., “kill people” vs. “kill time”), making disambiguation inherently more challenging. Despite this, G-SAE still provides meaningful improvements, albeit with smaller margins than PII.
>
> We outperform exclusively on all these various evaluations, which should naturally be the case due to the supervision, and have not found drawbacks yet. However, further specific suggestions are welcome, and we have added the discussion about concept ambiguity to the paper.
>
>
> **W3 (Running example):** We appreciate the feedback regarding our clear writing and welcome suggestions to make our work accessible to an even broader audience. As a short example, from the PII dataset of Tab. 8, in the vanilla SAE for the user-name “20jey.marlov”, both the Email and User Name features activate or none, whereas in G-SAE, only the User Name activates. The acc_0 score is low in the vanilla SAE, reflecting that the most predictive feature is not uniquely informative. Local disentanglement is also low, since removing the top feature has limited effect, as other features like Email still contribute overlapping signals. Similarly, global disentanglement is poor, as multiple features must be combined to reconstruct the concept effectively. In general, standard SAE activates with lower scores and is more ambiguous.
>
> In contrast, G-SAE has a strong single feature activation leading to high scores in acc_0, local and global disentanglement, resulting in a high FMS score.
>
> We will extend Tab. 8 to also display examples for the vanilla SAE activations and added this discussion to paragraph line 151.
>
>
> **Q1 (Used classifier):** Thank you for the insightful question. We chose binary decision trees primarily for their interpretability, sparsity, and alignment with the discrete, localized nature of monosemantic features. However, the FMS metric is classifier-agnostic by design: it evaluates concept capacity, local exclusivity, and global spread — all of which can be derived using any classifier that provides comparable accuracy signals.
> In theory, the choice of classifier should only marginally influence FMS, and mainly through differences in how classifiers optimize their objectives (e.g., Gini impurity for trees vs. hinge loss for SVMs). Some variation may occur depending on data sparsity and decision boundary shape, but the underlying structure of concept-feature alignment remains the same.
> As an example, we could compute FMS using SVMs in place of decision trees as follows:
> Acc_0: Train an SVM for each latent feature and take the feature with the highest accuracy as the best separating feature
> Local: Measure the accuracy drop between the highest and second-highest feature
> Global: Based on accuracies from 1. incrementally add features to be used in a new training of an SVM and document the increase in accuracy.
> This mirrors our decision tree-based implementation but substitutes SVMs as the base classifier. While the learned boundaries may differ slightly due to different optimization criteria, the structure and logic of FMS computation remain consistent, supporting its classifier-agnostic design.
> We have added an evaluation and discussion to the appendix.
>
>
> **Q2 (Cost of monosemanticity):** We further evaluated the trade-off between interpretability and general performance using SAE-Bench [1], a standard benchmark performing some specific tasks expected on SAE’s. As shown in the table below, G-SAE matches vanilla SAEs across all evaluation criteria, while being much more interpretable in our conditioned concept.
>
> This suggests that monosemanticity improvements do not come at the cost of model fidelity or general utility.
> SAE-Bench results (except for mse, higher is better):
> | dataset   | model       |   ce loss score |   mse |   mean absorption fraction score |   scr metric threshold 20 |   tpp threshold 50 total metric |   disentanglement score |   sparse probing top 1 |
> |:----------|:------------|----------------:|------:|---------------------------------:|--------------------------:|--------------------------------:|------------------------:|-----------------------:|
> | PII       | Vanilla SAE |           0.572 | 0.131 |                            0.061 |                     0.079 |                           0.219 |                   0.677 |                  0.740 |
> | PII       | G-SAE       |           0.688 | 0.264 |                            0.414 |                     0.074 |                           0.229 |                   0.685 |                  0.747 |
> | RTP       | Vanilla SAE |           0.814 | 0.328 |                            0.338 |                    -0.055 |                           0.276 |                   0.796 |                  0.766 |
> | RTP       | G-SAE       |           0.810 | 0.407 |                            0.607 |                     0.045 |                           0.310 |                   0.788 |                  0.754 |
> | SP        | Vanilla SAE |           0.604 | 0.351 |                            0.425 |                    -0.058 |                           0.194 |                   0.341 |                  0.738 |
> | SP        | G-SAE       |           0.692 | 0.350 |                            0.416 |                    -0.091 |                           0.251 |                   0.459 |                  0.745 |
>
>
>
> **Q3 (Ethical Concerns):** We have written a more extensive ethical impact statement in the appendix and the paper checklist, see Appendix A and Neurips Paper Checklist Questions 9,10, and 11. We agree that it deserves more attention and have moved it to the main body.
>
> [1] Karvonen et al., “SAEBench: A Comprehensive Benchmark for Sparse Autoencoders in Language Model Interpretability”, ICML (2025)

---

> > ### Comment · Reviewer_QSmM · 2025-08-04
> >
> > Thank you for the detailed responses to my concerns and questions. I am raising my score and think that your paper presents a solid contributions to the field with the added clarifications. Good job and hope to see your work at NeurIPS this year!

---

> ### Author Response · Authors · 2025-08-04
>
> Thanks for the response and raising the score! We appreciate the constructive feedback. If there is anything further we can help with, please let us know.

---

### Official Review · Reviewer_Agfr · 2025-07-03

**Clarity:** 4
**Significance:** 3
**Originality:** 3
**Rating:** 5
**Confidence:** 3

**Summary:**

This paper (a) proposes three evaluation metrics for characterizing the degree to which a given sparse autoencoder model (SAE) truly generates representations that semantically isolate a given concept. The paper then (b) proposes a loss modification to the SAE architecture that combines reconstruction loss (standardized) with a term that encourages one neuron to optimally predict the concept(s) of interest. The paper then evaluates the proposed method in the context of several exercises, ranging from toxicity detection in text to privacy concerns. The authors also discuss how the proposed architecture (g-SAE) can improve concept detection and controllable generation by intervening on the single, known latent neuron associated with the concept of interest (no expensive concept-discovery needed).

**Questions:**

What would a loss modification targeting global disentanglement look like?

**Ethical Concerns:**

["NO or VERY MINOR ethics concerns only"]

**Final Justification:**

This paper is well-written, identifies a problem with existing interpretability approaches, and proposes a simple solution. It has many strengths. One limitation is that the proposed method has higher data requirements at training time, which limits its use in data-scarce environments. I see no unresolved issues in my evaluation.

**Limitations:**

See discussion on local disentanglement in the modified loss.

An important limitation is that the original SAE work does not require labeled data for training. This g-SAE approach does. Thus, even though the g-SAE model would be expected to meet much better local disentanglement conditions than vanilla SAE for a given target concept, a given g-SAE cannot be applied to new scenarios. That said, if there is a strong scientific reason for measuring a given concept, the approach outlined here makes a lot of sense.

**Paper Formatting Concerns:**

The paper contains quite explicit content in Table 6. I would use a profanity filter (e.,g., ***).

**Quality:**

3

**Strengths And Weaknesses:**

--Strengths--

There is much to appreciate about this paper. The writing is exceptionally clear, the evaluation metrics are well-explained and useful, and the architecture change is well-motivated. The empirical evaluation is also solid. I find the simplicity of loss modification and the resulting ease of steering particularly attractive.

--Weaknesses--

The paper could provide a clearer explanation of where there is a need for both local and global measures of disentanglement. This is explained somewhat in the paragraph beginning, "Based on the notion above,..." but could be strengthened. The reader may also not gain a clear empirical sense of how correlated the two measures are for various model/concept pairs. It would be helpful to present some of these correlations. It would also help connect the metrics part of the paper with the loss modification part if the paper explained how the loss modification is more directly targeting improvements in local disentanglement. This could be perceived as a limitation (the loss only directly targets local disentanglement, although could improve both in practice).

---

> ### Author Rebuttal · Authors · 2025-07-31
>
> We appreciate that the reviewer finds our explanation very clear and easy to understand.
>
> In answer to your questions:
>
> **W (Global vs local measures):** We agree that additional examples can be even more helpful, and have added an example that better explains/distinguishes between the two forms. While related, they capture complementary aspects of monosemanticity:
> Local disentanglement measures exclusivity: How much of a concept is isolated in a single feature.
> Global disentanglement captures concept spread: Whether the concept is distributed across multiple features.
> Both are necessary:
> A feature may be predictive but redundant, i.e., multiple features encode the concept equally well. Leading to a high global score but low local disentanglement.
> Conversely, suppose a concept is primarily encoded in one feature but clearly improves when a few others are added. In that case, this yields higher local but lower global disentanglement, since the concept isn't entirely isolated.
> These distinctions justify evaluating both axes to measure monosemanticity robustly. We have updated our writing accordingly.
>
> **W (Empirical example):** We added a descriptive, easy-to-understand running example from the likes of Table 8, Appendix J.3, from the PII dataset. For example, in the vanilla SAE for the user-name “20jey.marlov”, both the Email and User Name features activate or none, whereas in G-SAE, only the User Name activates. The acc_0 score is low in the vanilla SAE, reflecting that the most predictive feature is not uniquely informative. Local disentanglement is also low, since removing the top feature has limited effect, as other features like Email still contribute overlapping signals. Similarly, global disentanglement is poor, as multiple features must be combined to reconstruct the concept effectively. In general, standard SAE activates little and more ambiguous. We added the example to the main text to the paragraph line 151.
> We will extend Tab. 8 as well to also display examples for the vanilla SAE activations and corresponding scores from FMS.
>
> **W (Global Disentanglement Loss):** As already pointed out, we do not, albeit possible, deploy an explicit global disentanglement loss. Nevertheless, we observed such good performance already, which is likely due to the shift in activation value ranges as seen in Fig. 4a: Through conditioning, the activation ranges naturally rise and would suppress the non-supervised concepts on activation. We follow up on this feedback in Q1 and add this discussion to Section 5.3.
>
> **Q (Loss modification for global disentanglement):** Thank you for the interesting question. Indeed, our current method directly targets local disentanglement by promoting isolated concept activation. While this additionally improves global disentanglement, as seen in Tab.2, we agree that it does not explicitly enforce it.
> A loss targeting global disentanglement likely involves an adversarial objective: encouraging one feature to encode the concept while explicitly suppressing activation in other features. However, this presents several design choices regarding:
> What signal should be used to "suppress" non-target features?
> How should overlapping or hierarchical concepts be handled?
> Should absent concepts be treated as zeros or smoothed labels (e.g., small values like 0.01)?
> We are actively exploring these directions and believe they merit a dedicated follow-up, but chose not to include them in this version due to the additional complexity. We now discuss this in the future work and outlook.
>
> **L (Generalizability of G-SAE):** Thank you for highlighting this important distinction. We would like to clarify that any practical use of SAEs for concept detection inevitably requires labeled data at some stage. Without labeled concept data, there is no meaningful way to interpret or validate the latent features discovered by an SAE.
> Existing approaches typically rely on post hoc analysis, e.g., probing classifiers [1,2], which use labeled data after training to identify and evaluate concepts. Our approach shifts this supervision to earlier: instead of using labels after training, we use them during training to guide the representation learning process directly. Importantly, this does not introduce new data requirements compared to conventional pipelines — the same labels are used, only at a different stage.
> In addition to the experiments of the paper, we have made an additional experiment demonstrating that the supervision of G-SAE can also be applied post hoc. Specifically, we took a conventional, pretrained SAE [3] that was initially trained without label supervision and then finetuned it using our G-SAE method. The fine-tuning cost is negligible compared to pretraining (a few million vs 8.5B). The experiment demonstrates that G-SAE (supervision) can also be applied successfully after SAE pretraining without losing the standard SAE performance, as shown in the Table below, which further demonstrates the usability of our approach. The general SAE ability remains similar (see ce loss score or mse) while the monosemanticity of the SAE improves significantly (see FMS scores).
> We added this discussion more thoroughly to our paper.
>
> | concept     | model type                  |   ce loss score |   mse |   Accuracy |   MS_global |   MS_local@1 |   MS_local@5 |   FMS@1 |   FMS@5 |
> |:------------|:----------------------------|----------------:|------:|-----------:|------------:|-------------:|-------------:|--------:|--------:|
> | Shakespeare | Pretrained                  |           0.991 | 0.002 |       0.81 |        0.87 |         0.04 |         0.16 |    0.37 |    0.42 |
> | Shakespeare | Finetuned [E:100, LR: 1e-5] |           0.977 | 0.003 |       0.86 |        0.92 |         0.15 |         0.22 |    0.46 |    0.49 |
> | Shakespeare | Finetuned [E:25,  LR: 1e-5] |           0.980 | 0.002 |       0.84 |        0.92 |         0.11 |         0.17 |    0.43 |    0.46 |
> | Shakespeare | Finetuned [E:100, LR: 1e-6] |           0.977 | 0.003 |       0.86 |        0.91 |         0.15 |         0.22 |    0.46 |    0.48 |
> | Toxicity    | Pretrained                  |           0.991 | 0.002 |       0.79 |        0.83 |         0.23 |         0.33 |    0.42 |    0.46 |
> | Toxicity    | Finetuned [E:100, LR: 1e-5] |           0.976 | 0.003 |       0.83 |        0.86 |         0.33 |         0.46 |    0.49 |    0.55 |
> | Toxicity    | Finetuned [E:25,  LR: 1e-5] |           0.977 | 0.003 |       0.83 |        0.86 |         0.33 |         0.45 |    0.50 |    0.55 |
> | Toxicity    | Finetuned [E:100, LR: 1e-6] |           0.975 | 0.003 |       0.83 |        0.87 |         0.33 |         0.44 |    0.50 |    0.54 |
>
> E=Finetuned Epochs; LR =Learning Rate
>
> **Formatting Concerns:** Thank you for the pointer. We agree and have applied the profanity filter to this content in our paper.
>
> [1] Bricken et al., “Towards Monosemanticity: Decomposing Language Models With Dictionary Learning”, Transformer Circuits Thread 2023
>
> [2] Gao, et al., "Scaling and evaluating sparse autoencoders.", ICLR (2025)
>
> [3] Hugginface: EleutherAI/sae-llama-3-8b-32x-v2 @ layer 11

---

> > ### Comment · Reviewer_Agfr · 2025-08-05
> > **Thanks to the Authors**
> >
> > Thanks to the Authors, for their comments here. These notes are helpful.
> >
> > I still think there is a difference in requiring data at the training stage vs. the evaluation stage (e.g., one team could train a single SAE and send the model to a large number of other teams, for different evaluations on different concepts), so this seems to strong: "Importantly, this does not introduce new data requirements compared to conventional pipelines — the same labels are used, only at a different stage." There is a "one-to-many"-ness of conventional SAEs that the proposed approach lacks - although it gains better disentanglement for a given target concept, as well noted.
> >
> > I have no remaining questions or comments; I retain my positive score.

---

> ### Author Response · Authors · 2025-08-05
>
> We appreciate the follow-up and thank the reviewer for the engagement and positive assessment.
>
> We agree our formulation in its current way is not fine granular enough. There is a difference between supervision during training versus applying it during evaluation, which we have integrated into this formulation.
> At the same time, our additional rebuttal experiment has shown that G-SAE supervision can be applied post-hoc to a pretrained SAE, suggesting this flexibility is not lost. Instead, it can be preserved through a lightweight fine-tuning step, adding concept-specific monosemanticity when needed, without discarding the benefits of general-purpose SAE pretraining.
>
> We have updated the paper to reflect this better. We hope to have addressed your concern and are happy about further discussion.

---

### Official Review · Reviewer_5oym · 2025-07-04

**Clarity:** 3
**Significance:** 3
**Originality:** 3
**Rating:** 4
**Confidence:** 4

**Summary:**

This paper proposes a novel metric, the Feature Monosemanticity Score (FMS), to measure the monosemanticity of features learned by sparse autoencoders (SAEs) in large language models (LLMs). The authors further introduce a new architecture called Guided Sparse Autoencoders (G-SAE), which incorporates supervision via a conditioning loss to improve monosemanticity and concept disentanglement. Empirical results demonstrate superior monosemanticity, detection, and steering capabilities compared to baseline SAE and state-of-the-art steering methods across tasks such as toxicity reduction, style modulation, and privacy preservation.

**Questions:**

Did you try any weighted loss function for the different components of loss function in G-SAE?

Please also check the weakness enlisted above.

**Ethical Concerns:**

["NO or VERY MINOR ethics concerns only"]

**Final Justification:**

This paper proposes a novel metric, the Feature Monosemanticity Score (FMS), to measure the monosemanticity of features learned by sparse autoencoders (SAEs) in large language models (LLMs).  I do not see any unresolved issue after rebuttal. I have already given positive score to this paper.  I would like to keep my score same.

**Limitations:**

yes

**Quality:**

3

**Strengths And Weaknesses:**

**Strengths**

 1. The proposed FMS metriccaptures not only the concept representation capacity of individual features but also their local and global disentanglement.

 2. G-SAE is a simple yet effective extension of SAEs that injects supervision during training, ensuring targeted features are better aligned with desired concepts.

**Weakness**

1.  For any kind of explainability a user study demonstrates the utility of the explanation.  For G-SAE a user study will enhance its contribution.

2. The motivation for using monosemanticity is not completely convincing. It would be nice to have one use case demonstrating the utility of monosemanticity for end user.

3. To apply G-SAE you would need a dataset where concepts are annotated. This kind of dataset may not always be available. This limits the use of the proposed approach.

---

> ### Author Rebuttal · Authors · 2025-07-31
>
> We thank Reviewer 5oym for their review, and appreciate that they find our introduced FMS metrics novel and discriminative, and our extension of SAE’s effective.
>
> In answer to your questions:
>
> **W1 (user study):** We respect the suggestion regarding a user study, which is indeed important for evaluating human perception and understanding of certain types of model explanations. However, G-SAE's core focus is on a more fundamental aspect: achieving directly measurable monosemanticity within latent features, meaning each feature cleanly represents a single, distinct concept. This property is rigorously and objectively measured via standard quantitative evaluation of features (e.g., acc_0) and our Feature Monosemanticity Score (FMS), e.g., Fig. 3 and 11, and can be directly compared to the labeled concept data. In this context, a user study would essentially parallel what our quantitative analysis already confirms. The utility of these monosemantic features is, moreover, clearly demonstrated by their successful deployment in practical applications such as toxicity control, privacy-aware generation, and style adaptation.
> We would appreciate suggestions for user studies that could further demonstrate and reinforce our findings.
>
> **W2 (motivation of monosemanticity):** We appreciate the reviewer’s perspective and agree that clearly demonstrating end-user relevance is important.
> To summarize, we show that G-SAE’s, through their monosemantic capability, merge the concept on a single neuron only. This enables us to use this single neuron in the entire model as a concept detector, which could be leveraged to the frontend as shown in Tab 8. Moreover, we show how using its corresponding output vector leads to an effective steering method. To realize these use cases in other methods [1,2,3] one needs to somehow merge the sparse and low activations, which may introduce further ambiguity.
> Section 5 demonstrates concrete use cases where monosemantic features directly benefit downstream utility, for detection as well as steering, such as toxicity reduction, writing style adaptation, and privacy preservation. Throughout the paper we follow this motivation and demonstrate with 26 distinct concepts that G-SAE exclusively outperforms all other compared SOTA methods.
>
> This demonstrates how monosemantic representations can enable more precise and user-relevant model behavior. We incorporated this discussion more prominently in the Introduction and Conclusion to further strengthen the need and relevance of monosemanticity.
>
> **W3 (Labeled data):** Thank you for highlighting this important distinction. We would like to clarify that any practical use of SAEs for concept detection inevitably requires labeled data at some stage. Without labeled concept data, there is no meaningful way to interpret or validate the latent features discovered by an SAE.
> Existing approaches typically rely on post hoc analysis, e.g., probing classifiers [4,5], which use labeled data after training to identify and evaluate concepts. Our approach shifts this supervision to earlier: instead of using labels after training, we use them during training to guide the representation learning process directly. Importantly, this does not introduce new data requirements compared to conventional pipelines — the same labels are used, only at a different stage.
> In addition to the experiments of the paper, we have made an additional experiment demonstrating that the supervision of G-SAE can also be applied post hoc. Specifically, we took a conventional, pretrained SAE [6] that was initially trained without label supervision and then finetuned it using our G-SAE method. The fine-tuning cost is negligible compared to pretraining (a few million vs 8.5B). The experiment demonstrates that G-SAE (supervision) can also be applied successfully after SAE pretraining without losing the standard SAE performance, as shown in the Table below, which further demonstrates the usability of our approach. The general SAE ability remains similar (see ce loss score or mse) while the monosemanticity of the SAE improves significantly (see FMS scores).
> We added this discussion more thoroughly to our paper.
>
> | concept     | model type                  |   ce loss score |   mse |   Accuracy |   MS_global |   MS_local@1 |   MS_local@5 |   FMS@1 |   FMS@5 |
> |:------------|:----------------------------|----------------:|------:|-----------:|------------:|-------------:|-------------:|--------:|--------:|
> | Shakespeare | Pretrained                  |           0.991 | 0.002 |       0.81 |        0.87 |         0.04 |         0.16 |    0.37 |    0.42 |
> | Shakespeare | Finetuned [E:100, LR: 1e-5] |           0.977 | 0.003 |       0.86 |        0.92 |         0.15 |         0.22 |    0.46 |    0.49 |
> | Shakespeare | Finetuned [E:25,  LR: 1e-5] |           0.980 | 0.002 |       0.84 |        0.92 |         0.11 |         0.17 |    0.43 |    0.46 |
> | Shakespeare | Finetuned [E:100, LR: 1e-6] |           0.977 | 0.003 |       0.86 |        0.91 |         0.15 |         0.22 |    0.46 |    0.48 |
> | Toxicity    | Pretrained                  |           0.991 | 0.002 |       0.79 |        0.83 |         0.23 |         0.33 |    0.42 |    0.46 |
> | Toxicity    | Finetuned [E:100, LR: 1e-5] |           0.976 | 0.003 |       0.83 |        0.86 |         0.33 |         0.46 |    0.49 |    0.55 |
> | Toxicity    | Finetuned [E:25,  LR: 1e-5] |           0.977 | 0.003 |       0.83 |        0.86 |         0.33 |         0.45 |    0.50 |    0.55 |
> | Toxicity    | Finetuned [E:100, LR: 1e-6] |           0.975 | 0.003 |       0.83 |        0.87 |         0.33 |         0.44 |    0.50 |    0.54 |
>
> E=Finetuned Epochs; LR =Learning Rate
>
> **Q1(Weighted loss function):** We explored weighted loss configurations, and our empirical findings indicated that excessive weighting of the conditioning term led to degraded reconstruction (i.e., G-SAE behaving like a probe). A 1:1 balance consistently gave the best trade-off between interpretability and fidelity across tasks. However, to further fine-tune, given a particular task at hand, it might be good to use a weight that is different than 1:1. We have added this investigation to the paper.
>
>
> [1] Liu et al. "In-context vectors: Making in context learning more effective and controllable through latent space steering.", arXiv preprint arXiv:2311.06668 (2023).
>
> [2] Rimsky et al. "Steering llama 2 via contrastive activation addition.", ACL (2024).
>
> [3] Subramani et al., "Extracting latent steering vectors from pretrained language models.", ACL (2022).
>
> [4] Bricken et al., “Towards Monosemanticity: Decomposing Language Models With Dictionary Learning”, Transformer Circuits Thread 2023
>
> [5] Gao, et al., "Scaling and evaluating sparse autoencoders.", ICLR (2025)
>
> [6] Hugginface: EleutherAI/sae-llama-3-8b-32x-v2 @ layer 11

---

> > ### Comment · Reviewer_5oym · 2025-08-03
> >
> > Thank you for your detailed response. I now find the justification regarding the labeled data convincing. However, I remain unconvinced by the explanations addressing Weaknesses 1 and 2. Therefore, I would  retain my original score.

---

> ### Author Response · Authors · 2025-08-04
>
> Thanks for your response and engagement! We are happy to clarify our claim regarding labeled data.
>
> We would like to clarify the other two points raised by the reviewer further.
>
> **Monosemanticity.** To further demonstrate the benefit of monosemanticity on cleaner detection and steering, we extended Tables 6, 7, 8 of Appendix J (SP, RTP as steering, PII as detection) with their corresponding vanilla-SAE samples and further annotated their measured FMS scores (average FMS@1 Vanilla SAE: 0.27 vs G-SAE 0.52, per-class details are added to the appendix). Some examples are shown in the tables below. In particular, in PII, it can be seen that vanilla frequently detects other concepts than the target concept. For steering into Shakespeare style, it often falls into repetition and does not obey the style appropriately.
>
> **User study.** To assess the human-centric alignment of our approach, we manually validated the F1 score against human-graded correctness (on a 0–100 scale, based on approximate percent matching) using 20 randomly selected samples. Participants evaluated the concepts detected by G-SAE against their own assessments. Results showed a strong positive correlation (R² = 0.85), indicating that the F1 score for G-SAE reliably reflects human judgment of correctness. Two independent annotators conducted the in-house evaluations. We are currently expanding this validation beyond the internal team, and final results will be included in the revised manuscript.
>
> We added these results and discussions to the paper and are looking forward to your feedback.
>
> ---
> Two parts of samples for tokenwise PII annotations taken from table 8 with respective maximal class activation values. In the first example, the vanilla SAE for the tokens of user-name “20jey.marlov” activated the Email, User Name and None, whereas in G-SAE, only the User Name activates.
>
> | Tokens:           | 20       | j        | ey       | .m       | al       | ov       |
> | ------------------ | -------- | -------- | -------- | -------- | -------- | -------- |
> | G-SAE class        | username | username | username | username | username | username |
> | G-SAE values       | .82      | 1.       | 1.       | 1.       | 1.       | 1.       |
> | Vanilla SAE class  | o        | username | username | email    | username | o        |
> | Vanilla SAE values | -        | .86      | .66      | .95      | .79      | -        |
>
>
>
>
> | Tokens:           | BACKGROUND | :   | 22     | :    | 41     | on   | December |      | 21   | st   | ,    |      | 196      | 6      |
> | ------------------ | ---------- | --- | ------ | ---- | ------ | ---- | -------- | ---- | ---- | ---- | ---- | ---- | -------- | ------ |
> | G-SAE class        | o          | o   | time   | time | time   | o    | date     | o    | date | date | o    | date | username | data   |
> | G-SAE value        | -          | -   | 1.     | 1.   | 1.     | -    | .6       | -    | .99  | .31  | -    | .86  | .44      | .93    |
> | Vanilla SAE class  | state      | o   | social | time | social | time | bod      | date | bod  | bod  | time | o    | social   | social |
> | Vanilla SAE values | .89        |   -  | .62    | .83  | 1.     | .98  | .75      | .71  | .98  | .82  | .52  |      | .99      | 1.     |
>
>
>
> ---
> Two steering examples for Shakespeare from table 7:
> |prompt|G-SAE|Vanilla SAE|
> |--|--|--|
> |After you cast your magic spell on me last time, I sent you a ring. | And then did you not stay, That you might for such use entertain'd be? And then to Sighing, with full thing of love, | I am now in the same town as you, and I am very anxious to hear from you. I am in good health and hope you are the same. |
> |Did she see you there the whole time, old boy? | And did you see her, and did see her goodly? And did her eyes, that so much grace doth give, And her sweet mouth, | I don't know. I was in the house. I was in the house. I was in the house. I was in the house. I was |

---

### Author Response · Authors · 2025-08-09
**Thanks to the reviewers**

We honestly appreciate the reviewer's feedback and engagement.

We are happy that the paper was positively received from the beginning, with reviewers noting the clarity of writing, the novelty of the FMS metric, the simplicity of the G-SAE method, and the strength of the empirical results.

During the rebuttal, we have added further experiments and clarifications, which led most reviewers to even increase their scores.

---

### Decision · Program_Chairs · 2025-09-17

**Decision:**

Accept (spotlight)

**Comment:**

This submission introduces a metric called the "Feature Monosemanticity Score" that quantifies the degree to which a feature encodes a single concept. Then, this new metric is used to design a variant of sparse autoencoders called G-SAEs that uses a labeled dataset of concepts to learn more monosemantic features.

The submission is strong, with clear writing, well-motivated and precise metrics, and a new architecture that optimizes for the proposed metrics. Theoretically and empirically the work is strong.

Reviewer cJ89 identified a weakness in that the authors did not include evaluations against standard metrics for evaluating SAEs, but the authors resolved these issues during rebuttal. Reviewer QSmM also raised concerns about the role of the labeled dataset in the G-SAE training, but the authors resolved these issues as well.

Ultimately, this is a strong submission that was further improved by the review process! I believe it is worth a spotlight.